# SD-MAD: SIGN-DRIVEN FEW-SHOT MULTI-ANOMALY DETECTION IN MEDICAL IMAGES

## ABSTRACT

Medical anomaly detection (AD) is crucial for early clinical intervention, yet it faces challenges due to limited access to high-quality medical imaging data, caused by privacy concerns and data silos. Few-shot learning has emerged as a promising approach to alleviate these limitations by leveraging the large-scale prior knowledge embedded in vision-language models (VLMs). Recent advancements in few-shot medical AD have treated normal and abnormal cases as a one-class classification problem, often overlooking the distinction among multiple anomaly categories. Thus, in this paper, we propose a framework tailored for few-shot medical anomaly detection in the scenario where the identification of multiple anomaly categories is required. We propose that separating anomalies relies on distinct radiological signs, routinely used by clinicians to bridge knowledge and images. To capture the detailed radiological signs of medical anomaly categories, our framework incorporates diverse textual descriptions for each category generated by a Large-Language model, under the assumption that different anomalies in medical images may share common radiological signs in each category. Specifically, we introduce SD-MAD, a two-stage **S**ign-**D**riven few-shot **M**ulti-**A**nomaly **D**etection framework: (i) Radiological signs are aligned with anomaly categories and distinguished by amplifying inter-anomaly discrepancy; (ii) Aligned signs are selected further to mitigate the effect of the under-fitting and uncertain-sample issue caused by limited medical data, employing an automatic sign selection strategy at inference. Moreover, we propose two protocols to comprehensively quantify the performance of multi-anomaly detection. Extensive experiments illustrate the effectiveness of our method.

## 1 INTRODUCTION

Medical anomaly detection (AD) has emerged as a critical area of research within the healthcare domain Fernando et al. (2021). The detection of anomalies, such as tumors Baid et al. (2021) and lesions Ding et al. (2022d), is essential for prompt clinical intervention. However, access to high-quality medical imaging data remains a significant challenge due to privacy concerns and institutional data silos, thereby highlighting the importance of few-shot learning approaches in medical anomaly detection.

Traditional few-shot anomaly detection Sheynin et al. (2021); Huang et al. (2022) often struggles to generalize the model from the limited data to a universal situation because of the limited prior knowledge scale of the model. Recently, many works Huang et al. (2024); Cao et al. (2024); Gu et al. (2024) utilize the large-scale vision-language model (VLM), such as CLIP Radford et al. (2021); Wang et al. (2022), to help improve the generalization ability of the model in medical anomaly detection. Similar to traditional anomaly detection methods, these approaches identify anomalies by designing a score function that determines whether a given input is normal or abnormal (one-class classification). However, in real-world scenarios, especially in medical imaging, it is crucial to distinguish between different categories of anomalies, as they may correspond to varying pathological conditions and require distinct clinical responses. For example, distinguishing between a lung tumor and pneumonia in chest X-rays is crucial, as they require different treatment approaches: surgery or chemotherapy for cancer Montagne et al. (2021), and antibiotics for infection Bassetti et al. (2022). Thus, this paper aims to investigate scenarios involving the presence of diverse anomaly types by

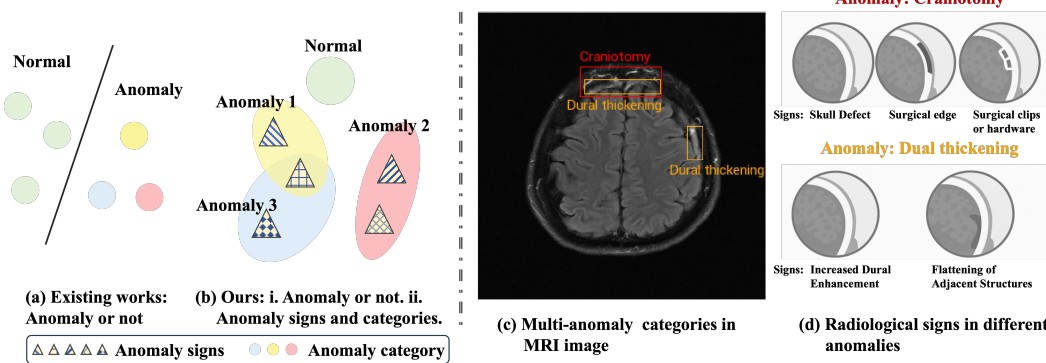

Figure 1: Figures (a) and (b) visualize the difference between our task and previous tasks. Figures (c) and (d) explain a multi-anomaly scenario, and radiological signs of different medical anomalies in the Brain MRI.

few-shot learning. The difference between the existing setting and our work is illustrated in Figure 1(a) and 1(b).

To explore the multi-anomaly scenario, we hypothesize that different anomalies in medical images may share common radiological signs, *e.g.*, ring-enhancing lesion can appear in pyogenic abscess, metastasis, and high-grade glioma. Furthermore, in each category, such as abnormal density or shape, while also exhibiting unique signs that are specific to each anomaly category. These distinct features can provide valuable diagnostic information, enabling more accurate classification and treatment planning. By leveraging both shared and unique patterns, we aim to improve the detection and distinction of various anomalies in medical imaging. Based on this hypothesis, firstly, we introduce a CLIP-based framework that explicitly **(i) links each anomaly class to a small set of textual "symptom" (signs) descriptions** and measures their similarity to image features. For each anomaly, we enumerate radiologic signs (e.g., "brain with craniotomy defect", "brain with unclear focal abnormality") as prompts. As shown in Figure 1(d), aligning visual embeddings with these sign prompts allows the model to learn fine-grained inter-anomaly distinctions. However, recent work Xia et al. (2024); Wang et al. (2022) reveals that prompt-based alignment in medical vision–language models can be uncertain: not all signs contribute equally, and some may even introduce noise in intra-class matching. To address this, at inference time, we **(ii) automatically select the most informative prompts for each few-shot example** Shum et al. (2023), thereby mitigating misleading matches within the same anomaly class. By addressing both inter-anomaly and intra-anomaly challenges, our approach delivers more accurate and reliable multi-anomaly detection under few-shot conditions.

We structure the evaluation protocol for the multi-category medical AD task around three layers to capture the full spectrum of multi-anomaly detection performance: (1) assessing the model's ability to distinguish between normal and abnormal instances; (2) evaluating the model's ability to perform multi-label prediction across distinct anomaly types; and (3) assessing the model's ability to correctly identify the specific types of anomalies. Existing methods face challenges in adapting to the last two protocols, primarily because their scoring functions are not designed to generalize to these task settings.

As summarized below, our contributions are threefold:

1. **Framework for few-shot multi-anomaly detection.** We introduce a few-shot anomaly detector that natively handles multiple anomaly classes within a single model, based on learning the alignment of radiological signs and anomaly categories.

2. **Inter- and intra-anomaly alignment.** We align image embeddings with sets of anomaly-specific prompts during training and, at inference, automatically select the most informative prompts to mitigate the uncertain-sample issue in the vision–language alignment.

3. **Rigorous evaluation protocol.** We assess our approach on three medical imaging datasets across the evaluation protocols of medical multi-anomaly detection. Moreover, we evaluate SD-MAD on the general medical anomaly detection benchmark which contains 6 datasets

for evaluation(see the Appendix). The results demonstrate consistent improvements over state-of-the-art baselines.

## 2 RELATED WORK

**Medical Anomaly detection.** Traditional medical anomaly detection methods rely on well-curated anomaly datasets, training on normal images and evaluating on abnormal ones Bao et al. (2024); Cai et al. (2023); Zhang et al. (2020); Zhou et al. (2021); Xiang et al. (2023); Hassanaly et al. (2024); Linmans et al. (2024); Graham et al. (2023). These approaches model the normal data distribution and identify anomalies as deviations from this distribution, achieving impressive performance. Many of these methods are designed for specific anatomical regions Ding et al. (2022c); Xu et al. (2024) and treat anomaly detection (AD) as a one-class classification problem Bao et al. (2024); Cai et al. (2023); Jiang et al. (2023). However, in real-world scenarios, the same individual may experience multiple diseases affecting the same organ. Recently, the open-set AD method Zhu et al. (2024) has shifted focus to detecting multiple anomalies instead of relying on one-class classification. These methods require enough training data to formulate the expected distributions, which can be hard to adapt to few-shot setting. To address the challenge of limited large-scale labeled datasets, some approaches have explored few-shot anomaly detection techniques as follows.

**Few-shot Anomaly detection.** Few-shot anomaly detection has gained significant attention in recent years due to its ability to identify rare or unseen anomalies with limited labeled data. Previous models utilized disentangled representations of anomalies Ding et al. (2022a) or contrastive learning mechanisms Yao et al. (2023a) to alleviate the bias, accounting for unseen anomalies. MVFA Huang et al. (2024) utilized multi-level adaptation and a contrastive framework to improve generalization across various medical datasets. UniVAD Gu et al. (2024) proposed a general framework to detect anomalies across different domains with a training-free unified model. AA-CLIP Ma et al. (2025) advanced CLIP model in a two-stage approach to enhance CLIP's anomaly discrimination ability. Although those methods perform well in various datasets, there is still a lack of few-shot multi-anomaly detection for medical data.

**Vision-language model.** Vision-language models have demonstrated significant potential across a range of tasks. CLIP Radford et al. (2021) excels in image-text alignment and has been successfully applied to various applications, such as classification and text-image retrieval. To expand CLIP's capabilities to medical data, MedCLIP Wang et al. (2022) was introduced as a foundation for medical image-text alignment. Based on those pre-trained foundation models, recent studies Hua et al. (2025); Jin et al. (2025); Cao et al. (2025) in anomaly detection have leveraged pre-trained CLIP models for language-guided anomaly detection and segmentation, achieving impressive results and highlighting the promising potential of these models in this domain.

## 3 METHODOLOGY

In this section, we first formulate the problem of few-shot anomaly detection and few-shot multi-anomaly detection in medical images. Then we propose our methods within two parts: In section 3.2, we propose a training method with a tailored adapter for vision-language models and an inter-anomaly representation learning loss function; In Section 3.3, we propose an inference strategy to filter the outlier prompts, which aims to handle the intra-anomaly uncertain samples. Figure 2 shows the overall pipeline of our model.

### 3.1 PROBLEM FORMULATION

**Few-shot medical anomaly detection**: Following the setting of previous work Huang et al. (2024) on few-shot medical anomaly detection, the few-shot training samples can be presented as $\mathcal{D}_{few} = \{(x_i, c_i, s_i)\}_i^K$,

where $K$ is the number of samples, $x_i$ is the $i$-th image, the corresponding image-level label $c_i \in \{0, 1\}$, and the pixel-level label $s_i \in \{0, 1\}^{h \times w}$ is a binary mask with the same size $h \times w$ as the image $x_i$. For a given test image $x_{test}$, image-level and pixel-level medical anomaly detection are evaluated with the corresponding image labels $c_{test}$ and pixel labels $s_{test}$.

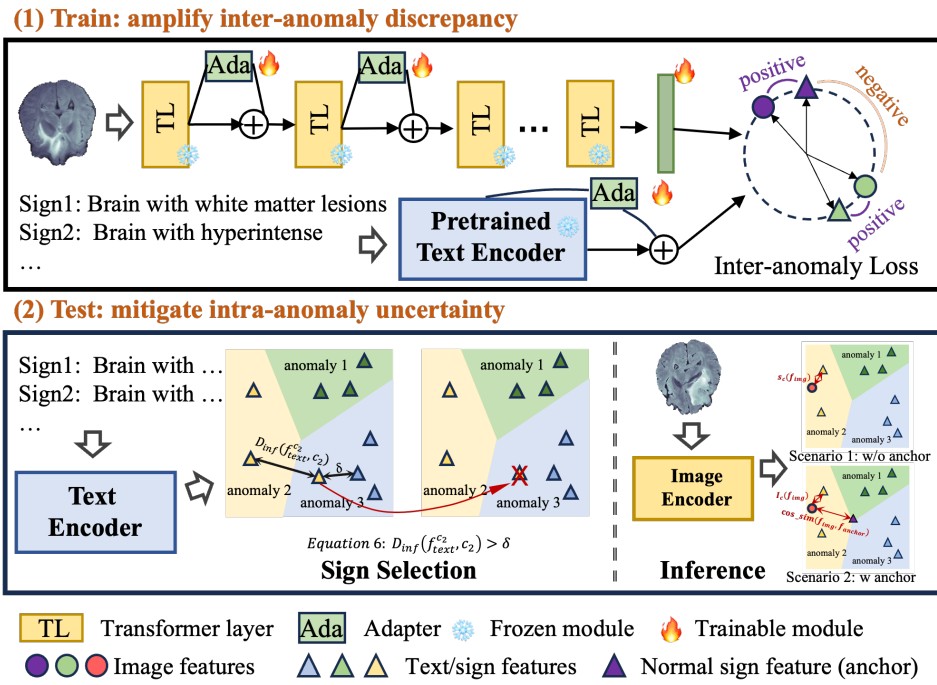

Figure 2: The pipeline of SD-MAD. In the framework, the training phase is designed to amplify inter-anomaly discrepancies, and the inference stage aims to handle the uncertain-sample problem in each anomaly category.

**Few-shot medical anomaly detection with multiple anomaly categories**: Similar to the setting of few-shot medical anomaly detection, few-shot training samples can be presented as $\hat{\mathcal{D}}_{few} = \{(x_i, \mathbf{c}_i)\}_i^K$, where $\mathbf{c}_i \in \{0, 1\}^d$ is a $d$-dimensional label. Since it is hard to access the pixel-level labels for the multi-anomaly medical datasets, we do not consider the pixel-level label in this setting. Thus, given a test image $x_{test}$, only image-level medical anomaly detection is evaluated with the corresponding image labels $\mathbf{c}_{test}$ in the scenarios where multiple anomaly categories exist.

## 3.2 TRAINING: AMPLIFY INTER-ANOMALY DISCREPANCY

**Shift Adapter.** To preserve the large-scale prior knowledge encoded in CLIP, we propose a shift adapter designed to effectively aggregate learning signals from few-shot samples while retaining the original prior information. The shift adapter is used for both image and text encoders, which is shown as our pipeline in Figure 2.

Considering the feature $\hat{f}_i^{in}$ is input of the adapter, which is also the output of the $i$-th transformer layer, the output of the adapter at the $i$-th transformer layer is

$$\hat{f}_i^{ada} = \alpha(W_i^2 \alpha(W_i^1 \hat{f}_i^{in})), \tag{1}$$

where $W_i^1$ and $W_i^2$ are trainable linear weights of the adapter at the $i$-th transformer layer, $\alpha$ is the activation function.

Inspired from residual learning methods He et al. (2016), we integrate the output of the original transformer layer $\hat{f}_i^{out}$ with $\hat{f}_i^{ada}$ by inner interpolation as follows:

$$f_i^{out} = \lambda \hat{f}_i^{out} + (1 - \lambda) \hat{f}_i^{ada}, \tag{2}$$

where $\lambda$ is the hyperparameter to control the interpolation ratio. To avoid the overfitting caused by the limited number of few-shot samples, we restrict the application of the adapter to four layers in the image encoder and one layer in the text encoder.

**Inter-anomaly Loss.** The text-vision alignment in CLIP depends on this contrastive learning insight with the cosine similarity. From the view of contrastive learning Oord et al. (2018); Schroff et al. (2015), the distance of the positive image-text pairs should be smaller than the distance of negative text-image pairs. Towards this end, existing work directly minimize the cosine distance between the positive image-text pairs to align the text and image features as follows:

$$L_{img-text} = \min_{\theta} \sum_{i \in [1, N_c]} d(f_{img}^c, f_{text,i}^c).$$  (3)

Here, $d(\cdot, \cdot)$ denotes the cosine distance between two input vectors, $\theta$ is the trainable parameters, image feature $f_{img}^c$ and detailed-description text features $f_{text,i}^c$ belong to anomaly category $c \in \mathcal{C}$ of the given image, $N_c$ is the number of text prompts corresponding to category $c$.

It is important to note that Equation 3 does not account for the distances of negative pairs. This is because simply increasing the distance between negative pairs provides limited utility in enabling the model to accurately identify the anomaly categories. For instance, given an abnormal image exhibiting only the anomaly of a lesion, the prediction may still fail despite strong alignment of positive pairs, as the model may erroneously assign high similarity scores to irrelevant categories, resulting in false positives. To handle this issue, we introduce an anchor feature $f_{anchor}$ that serves to define the boundary between normal and abnormal images. Thus, the following relationship should be satisfied.

**Remark 3.1** *Given an image feature belonging to category c, we have*

$$\sup_{i \in [1, N_c]} d(f_{img}^c, f_{text,i}^c) \leq d(f_{img}^c, f_{anchor}) \leq \inf_{k \neq c, j \in [1, N_k]} d(f_{img}^c, f_{text,j}^k)$$

Given the image feature $f_{img}^c$ and category $c$, Remark 3.1 indicates that $f_{anchor}$ serves as the hyperplane to separate the subspace of category $c$ and other categories. To distinguish the difference between the normal category and other anomalies simultaneously, we set the $f_{anchor}$ as the feature of the text prompt corresponding to normal images. According to Remark 3.1, we propose the following loss:

$$\hat{d}_{positive,i}^c = \max(0, d(f_{img}^c, f_{text,i}^c) - d(f_{img}^c, f_{anchor}))$$

$$\hat{d}_{negative,j}^{c,k} = \max(0, d(f_{img}^c, f_{anchor}) - d(f_{img}^c, f_{text,j}^k))$$

$$L_{anchor} = \min_{\theta} \sum_{i \in [1, N_c]} \hat{d}_{positive,i}^c + \sum_{k \neq c, j \in [1, N_k]} \hat{d}_{negative,j}^{c,k}$$  (4)

As discussed above, the overall loss for amplifying the inter-anomaly discrepancy is

$$L = L_{img-text} + L_{anchor}$$  (5)

### 3.3 INFERENCE: MITIGATE INTRA-ANOMALY UNCERTAIN-SAMPLE ISSUE

During the inference stage, image features from the test set are evaluated against the text prompt features corresponding to each anomaly category. However, the limited number of few-shot training samples, combined with uncertainty in medical vision-language Xia et al. (2024), may cause underfitted features that fail to capture anomaly characteristic-specific information. Thus, to address this issue, we divided our inference stage into two parts as follows.

**Sign Selection.** As we discussed above, each anomaly category contains several prompt features corresponding to the anomaly signs. Thus, there should be a labeling function $h_{text}(\cdot)$ satisfied $h_{text}(f_{text}^c) = c$. Therefore, we have the definition of the distance between given text feature $f_{text}$ and category $c$ in the following.

**Definition 3.2** *Given a text feature $f_{text}$, the distance between the prompt feature $f_{text}$ and the decision region of the anomaly category c is*

$$D_{inf}(f_{text}, c) = \inf_{\{f'_{text} | h(f'_{text}) = c, f'_{text} \neq f_{text}\}} d(f'_{text}, f_{text})$$

Definition 3.2 provides a definition of distance between the prompt feature $f_{text}$ and the decision region $\{f'_{text}|h(f'_{text}) = c\}$. For the ideal situation, we have the following relation.

**Remark 3.3** *Given a text feature $f^c_{text}$ belonging to category $c$, we have*

$$D_{inf}(f^c_{text}, c) < \delta$$

*Where $\delta \triangleq \inf_{k \neq c, k \in \mathcal{C}} D_{inf}(f^c_{text}, k)$*

As shown in Figure 2, the outlier text features in each category may break the relation in Remark 3.3. However, in the inference time, the text features are fixed. Thus, we propose to modify the labeling function $h(\cdot)$ to mitigate this problem.

Given a text feature $f^c_{text}$ which satisfies $h(f^c_{text}) = c$, the new labeling function is defined as

$$h_{new}(f^c_{text}) = \begin{cases} c & \text{if } D_{inf}(f^c_{text}, c) < \delta, \\ -1 & \text{else.} \end{cases} \quad (6)$$

The Equation 6 indicates that the new labeling function $h_{new}(\cdot)$ discards the distorted features that break the relation in Remark 3.3 for each anomaly category. The sign selection process can be viewed in Fig. 2 (2). This labeling function is used for the score function design, which we will discuss in the following.

**Inference.** Unlike previous methods Huang et al. (2024); Jeong et al. (2023), which focus solely on evaluating the Area Under the Receiver Operating Characteristic curve (AUROC) using a continuous scoring function, we additionally consider scenarios that require binary predictions. For the binary prediction, the anchor feature is required for the evaluation.

**Scenario 1: Continuous scoring function without anchor feature.** Without anchor feature, for the given category $c$ and the image feature $f_{img}$, the score function corresponding to $c$ is

$$s_c(f_{img}) = \sup_{h_{new}(f_{text})=c} cosine\_similarity(f_{img}, f_{text}). \quad (7)$$

As we discussed in Section 3.1, the label of image $x$ is a vector $\mathbf{c}$. Thus, the score vector corresponding to $\mathbf{c}$ is $\mathbf{s_c} = \{s_{c_i}(f_{img})\}^K_{i=1}$, where $K$ is the number of anomaly categories.

**Scenario 2: Binary prediction with anchor feature.** With the anchor feature, we can achieve the binary prediction for each anomaly category. The prediction $p_c$ for category $c$ with a give image feature $f_{img}$ is

$$p_c(f_{img}) = \begin{cases} 1 & \text{if } I_c(f_{img}) > cosine\_similarity(f_{img}, f_{anchor}), \\ 0 & \text{else.} \end{cases} \quad (8)$$

, where $I_c(f_{img}) \triangleq \inf_{h_{new}(f_{text})=c} cosine\_sim(f_{img}, f_{text})$. The precision vector corresponding to $\mathbf{c}$ is $\mathbf{p_c} = \{p_{c_i}(f_{img})\}^K_{i=1}$. This prediction can be used for the evaluation with the Hamming score and the subset accuracy score.

## 4 EXPERIMENTS

### 4.1 EXPERIMENTAL SETUP

**Evaluation protocols** We introduce three evaluation protocols: 1) for general anomaly detection, following previous works Huang et al. (2024); Jeong et al. (2023), we quantify the performance with area under the receiver operating curve (AUROC) metric on image- and pixel-level; 2)We introduce Hamming score and subset accuracy to evaluate the performance on multi-label prediction on the task of multi-anomaly detection; 3) We exploit the AUROC metric for each class to evaluate the performance on the specific types of anomalies. Specifically, given the anomaly type $c$, the binary label is set as 1 for the images belonging to type $c$, and 0 for the others.

For general medical few-shot anomaly detection, we follow the benchmark setting of previous few-shot medical anomaly detection Huang et al. (2024); Bao et al. (2024). Since the general medical does not align with our principal research target, we report the experimental results in the appendix.

For multi-anomaly medical few-shot medical anomaly detection, we introduce two evaluation protocols: **Integrated Anomaly Detection:** To evaluate the model's ability to perform multi-label prediction across distinct anomaly types, we introduce the Hamming score and subset accuracy as the evaluation metrics. The detail of the two metrics is presented in the Appendix.; **Anomaly-wise Detection:** To assess the model's ability to correctly identify the specific types of anomalies, we introduce the anomaly-aware AUROC (Area Under the Receiver Operating Characteristic Curve) as the evaluation metric in this protocol.

As previous few-shot anomaly detection methods can not handle the multi-category scenarios, we only compare the baseline model CLIP Radford et al. (2021) and the vision-language model tailored for medical image MedCLIP Wang et al. (2022). We conduct the one-shot setting for the multi-anomaly detection, aligning with the practical constraint that rare-disease data are difficult to obtain.

**Dataset** The experiments on multi-anomaly detection are built from the 3 datasets: ChestX-ray8Wang et al. (2017b), OCT17 Kermany et al. (2018), and brain data in fastMRI+ Zhao et al. (2022); Zbontar et al. (2018). ChestXray8 contains 14 anomalies in chest X-ray images, and OCT-17 contains 3 anomalies in retinal optical coherence tomography images. For brain MRI data in FastMRI+, we select 6 anomaly categories and the same slice-level images, namely slice 0, 5 and 10, for the multi-anomaly detection tasks. We visualize different slices in the Appendix.

**Training details** We select CLIP with ViT-L/14 Dosovitskiy et al. (2020) as the backbone model with the size of input as $240\times240$. We employ our shift adapter to the 6-,8-,18- and 24-th layers in the transformer of the CLIP image encoder and to the last layer to the transformer of the CLIP text encoder. Every training process is conducted in 50 epochs. The training process requires 4000 Mib GPU memory for the model. The experiments are conducted on an A100 GPU.

## 4.2 EXPERIMENTS ON CHESTX-RAY8

We conduct the multi-anomaly-detection under two settings, namely Integrated Anomaly Detection and Anomaly-aware Detection. The experimental results are shown in Table 1. The detailed experiments for Anomaly-aware Detection can be viewed in the Appendix.

As shown in Table 1, our method significantly improves the performance of the CLIP model. Particularly, our training strategy contributes most to the improvements. In addition, MedCLIP performs better than the original CLIP, suggesting that incorporating domain-specific prior knowledge in multi-modal pretraining is meaningful for performance improvement in downstream tasks.

Table 1: The multi-anomaly detection results of the experiments on ChestX-ray8. The first row displays the results under the protocol of integrated anomaly detection, with an averaged AUROC score (Avg. AUROC). The category-wise results are provided in the Appendix. The last two rows display the results under the anomaly-aware detection protocol with Hamming score and subsect accuracy(Subset Acc.)."SS" is short for "Sign Selection".

|  | CLIP | MedCLIP | Ours (no SS) | Ours (full model) |
|---|---|---|---|---|
| **Avg. AUROC(%)** | 48.3 | 52.4 | 56.0 | **57.5** |
| **Hamming score(%)** | 4.4 | 18.3 | 95.2 | **95.6** |
| **Subset Acc.(%)** | 0 | 0.1 | 56.2 | **59.1** |

## 4.3 EXPERIMENTS ON OCT-17

We conduct the experiments under the Anomaly-aware Detection setting on OCT-17 dataset, which is shown in Table 2. As each image in OCT-17 exhibits only one anomaly class, co-occurrence does not arise. Accordingly, the Anomaly-aware Detection setting is adequate to assess multi-anomaly detection performance on OCT-17.

As shown in Table 2, our method can significantly improve the performance of multi-anomaly detection on average. The drop in performance on the anomaly CNV without sign-selection demonstrates the uncertainty problem in the training process. Our sign-selection procedure mitigates this issue by discarding outlier signs. This can, however, reduce performance in certain anomaly, e.g., DME,

Table 2: The experimental results under anomaly-aware detection protocol on OCT-17 dataset. We report the results over 3 anomalies, namely CNV(Choroidal Neo-vascularization, DME(Diabetic Macular Edema), and Drusen. Avg. is short for average."SS" is short for "Sign Selection".

| Anomaly | CLIP | MedCLIP | Ours (no SS) | Ours (full model) |
|---|---|---|---|---|
| CNV | 70.8 | 58.8 | 61.4 | 91.7 |
| DME | 60.3 | 36.6 | 81.5 | 71.3 |
| Drusen | 32.4 | 64.4 | 60.1 | 79.5 |
| **Avg.** | 54.5 | 53.3 | 67.7 | **80.8** |

where discarded signs may help to identify negative samples. Nevertheless, the method outperforms competing models on average.

## 4.4 EXPERIMENTS ON FASTMRI+

We evaluate the generalization ability of our model under different scenarios of brain MRI images in FastMRI+. Specifically, we evaluate the performance in different slices, namely slices 0,5, and 10. We demonstrate our experimental results under two multi-anomaly detection protocols in the following.

Table 3: The experimental results under the Integrated Anomaly Detection protocol on FastMRI+. The evaluation metrics are Hamming score and subset accuracy. "SS" is short for "Sign Selection"

|  |  | Clip | MedClip | Ours (no SS) | Ours (full model) |
|---|---|---|---|---|---|
| slice 0 | Hamming(%)↑ | 80.2 | 77.1 | 85.8 | **87.2** |
|  | Subset acc.(%)↑ | 0.4 | 18.5 | 34.6 | **60.8** |
| slice 5 | Hamming(%)↑ | **77.6** | 63.5 | 72.7 | 76.5 |
|  | Subset acc.(%) | 0 | 0 | **29.0** | 27.3 |
| slice10 | Hamming(%)↑ | 78.3 | 73.2 | 73.8 | **79.2** |
|  | Subset acc.(%)↑ | 0 | 1.9 | 19.8 | **21.7** |

**Integrated Anomaly Detection** Table 3 shows the results under the protocol of integrated anomaly detection on FastMRI+ dataset. As shown in the table, our full model demonstrates superior performance compared to the other methods in most cases. In slices 0 and 10, our method consistently outperforms other models. We find that the good performance on CLIP in slice 5 is attributed to excellent recognition of craniotomy, which is clearer and easier to recognize in slice 5. The detailed Anomaly-aware detection in the Appendix can also illustrate this phenomenon. Our method balances the decision region across anomalies, which leads to a modest performance drop.

**Anomaly-aware Detection** As Table 4 shows, our methods significantly improve the average performance for every slice. The consistent improvement over different slices also illustrates the generalization ability of our method under the Anomaly-aware detection protocol.

Table 4: The experimental results under the Anomaly-aware Detection protocol on FastMRI+. We report the average AUROC for each slice. Detailed results for each anomaly can be viewed in the Appendix. "SS" is short for "Sign Selection".

|  | CLIP | MedCLIP | Ours(no SS) | Ours(full model) |
|---|---|---|---|---|
| Slice | 57.5 | 50.3 | 67.3 | **68.2** |
| Slice 5 | 60.3 | 51.6 | 63.0 | **63.9** |
| Slice 10 | 60.5 | 47.1 | **67.1** | 61.5 |

## 4.5 VISUALIZATION OF IMAGE-TEXT SIMILARITY

To evaluate the alignment of our method, we visualize the alignment in Figure 3. As the figure shows, our training method promotes the alignment between abnormal images and corresponding prompts. However, some prompts may exhibit overconfidence in a false category. For instance, in Figure

3b, the characteristic *surgical scaring* has equally high similarities between both *Craniotomy* and *Posttreatment change* images. This phenomenon may result in uncertain samples for the prediction, which leads us to propose the sign selection method.

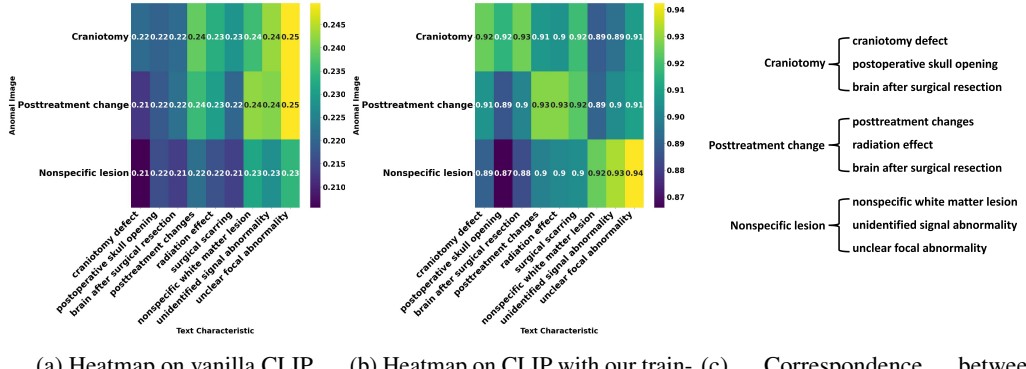

(a) Heatmap on vanilla CLIP    (b) Heatmap on CLIP with our training method    (c) Correspondence between prompts and anomaly categories

Figure 3: Visualization of image-text similarity heatmaps. (a) visualizes the heatmap on vanilla CLIP, (b) visualizes the heatmap on our trained model. The correspondence between prompts and anomaly categories is provided in (c).

## 4.6 ABLATION STUDY ON $\lambda$

To evaluate the effect of the hyperparameter in the Shift Adapter, we conducted ablation studies for $\lambda$ on the multi-label prediction with the 5th slice. Figure 4 shows the experimental results. The figure does not exhibit significant impacts on the performance when $\lambda$ changes, which shows the robustness of the learnable adapter. In addition, we find that there is a trade-off between Hamming score and Subset accuracy when $\lambda$ increases. We assume that it is because the increase of $\lambda$ may cause slight overfitting to the few-shot samples. Therefore, the model may produce fewer predictions in the presence of intra-class variation within the same anomaly type. While this may lead to a reduction in Hamming score, it could potentially enhance the overall prediction accuracy.

## 5 CONCLUSION AND LIMITATION

In this paper, we introduce a novel setting for medical anomaly detection, termed multi-anomaly detection. Unlike previous settings that typically assume a single anomaly per image, multi-anomaly detection is designed to address scenarios where multiple anomalies co-exist within the same clinical image. Building on this new task, we propose a method based on a vision-language model (VLM) for both inter- and intra-anomaly alignment. Specifically, we propose an inter-anomaly loss to amplify the inter-anomaly discrepancy and update the CLIP model with trainable Shift Adapters. In addition, we design a sign selec-

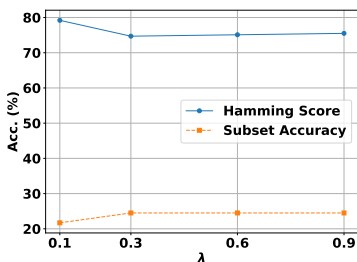

Figure 4: The ablation study on $\lambda$. We conduct the experiments on the multi-label prediction task with two metrics, namely Hamming score and Subset accuracy.

tion method to mitigate the intra-anomaly uncertainty at the inference stage. To thoroughly evaluate the performance of our method in the task of multi-anomaly detection, besides the general setting in anomaly detection, we propose two more evaluation protocols, namely multi-label prediction and category-wise AUROC. The extensive experiments illustrate the effectiveness of our method.

**Limitation** Even if our proposed method can effectively address the multi-anomaly detection task, there are still limitations, which mainly rely on the correspondence between the prompt and the anomaly categories. Some prompts may correspond to more than one anomaly type, which may result in false predictions if we ignore this nature. Addressing this ambiguity in prompt-anomaly correspondence will be the focus of our future work.

ETHICS STATEMENT

This work fully complies with the ICLR Code of Ethics. All research activities were performed with honesty, transparency, and a commitment to reproducibility. Experimental results are truthfully presented. No personal, sensitive, or proprietary data were used, and there are no privacy or licensing concerns associated with this study. No foreseeable harm arises from the work, prior literature and funding sources are cited, and there are no conflicts of interest. The paper does not infringe on institutional, legal, or ICLR ethical standards.

REPRODUCIBILITY STATEMENT

Significant efforts were made to ensure the reproducibility of all findings. The main body includes thorough descriptions of each model, training routine, hyperparameter setting, and evaluation strategy. Details of experiments are also provided in the appendix. All datasets are openly accessible.

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

## A APPENDIX 1: LLM USAGE

Large language models (LLMs) were used only for minor editorial tasks, such as refining grammar and improving linguistic clarity. Moreover, LLMs are also utilized during the procedure of generating candidate physiological signs.

## B APPENDIX 2: DETAILED DESCRIPTION OF MULTI-ANOMALY DETECTION DATASET

We adopt the Brain MRI dataset in FastMRI Zhao et al. (2022); Zbontar et al. (2018). The dataset consists of 3D Brain MRI models with 16 slices in each model. Each slice can be presented as an image. Figure 4 visualizes the 16 slices of a normal sample in FastMRI. We select slices 0, 5, and 10 to show the performance of the methods over different parts of Brain MRI.

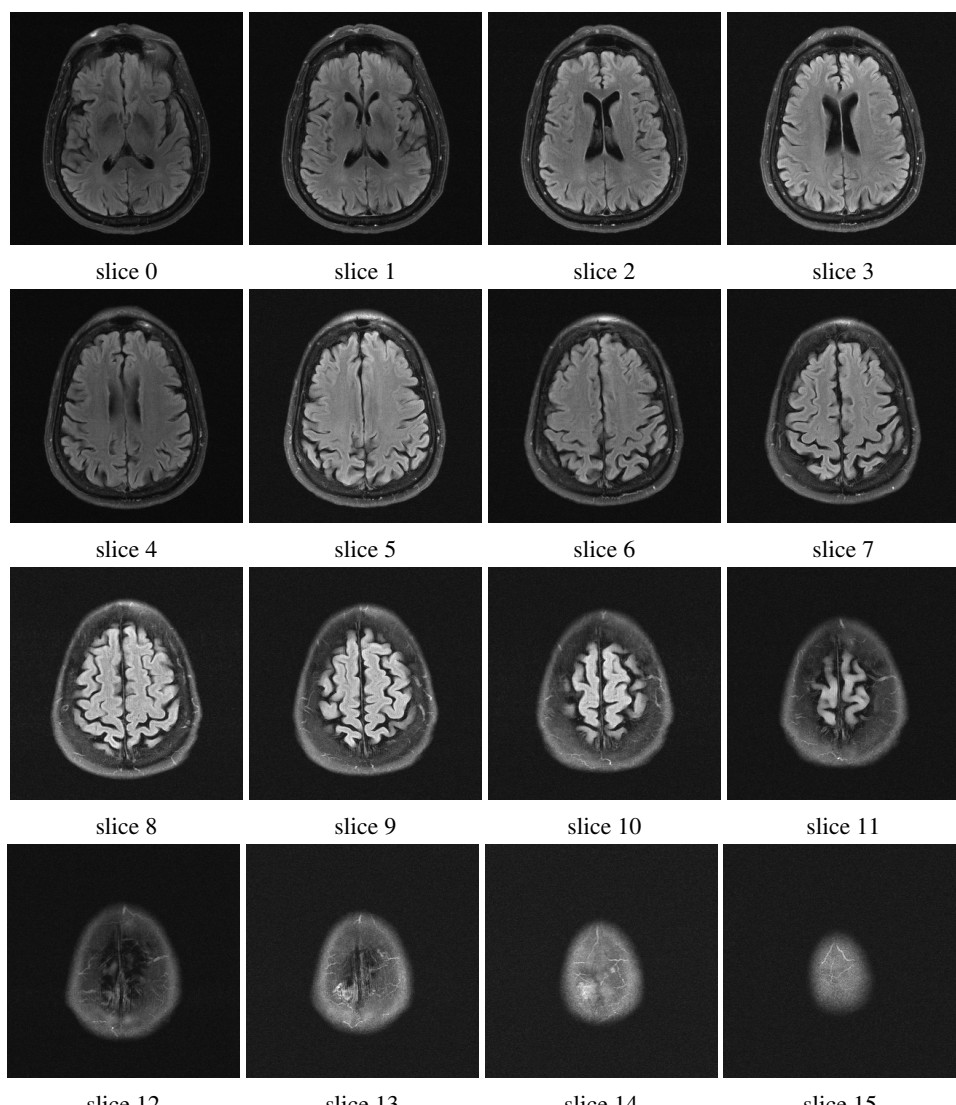

Figure 5: Visualization of data sample in FastMRI

## C  APPENDIX 3: EVALUATION METRICS

The Hamming score is calculated as follows:

$$Hamming\_score = 1 - \frac{1}{NK} \sum_{i=1}^{N} \sum_{j=1}^{K} I(p_j^i \neq c_j^i)$$

where $N$ is the number of samples, $K$ is the number of different categories, $p_i^j$ is the predication for the $j$-th category in the $i$-th sample image, $c_j^i$ is the ground-truth label of the $j$-th category in the $i$-th sample and $I$ is the characteristic function. The introduced Hamming score indicates the proportion of incorrect predictions compared with all predictions. Obviously, the larger the Hamming score is, the smaller the proportion is.

To measure the full correctness of the prediction, we also introduce the subset accuracy as follows:

$$Subset\_accuracy = \frac{1}{N} \sum_{i=1}^{N} I(\mathbf{p}^i = \mathbf{c}^i)$$

Where $N$ is the number of samples, $\mathbf{p}^i$ is the prediction vector of the $i$-th sample, $\mathbf{c}^i$ is the ground truth label vector of the $i$-th sample and $I$ is the characteristic function. The subset accuracy measures the proportion of samples for which the entire predicted label vector exactly matches the true label vector. This is a more restricted metric, as even a single incorrectly predicted label causes a zero value for the corresponding sample.

## D    APPENDIX 4: GENERAL FEW-SHOT MEDICAL ANOMALY DETECTION

Table 5: Comparison on general anomaly detection. "Avg." is short for "average".

| | Dataset | DRA | BGAD | MVFA | WinCLIP | AnomalyCLIP | InCTRL | Ours |
|---|---|---|---|---|---|---|---|---|
| | BrainMRI | 80.6 | 83.6 | 92.4 | 66.8 | 90.9 | 89.5 | 91.4 |
| | LiverCT | 59.6 | 72.5 | 81.2 | 67.2 | 92.4 | 90.1 | 86.9 |
| Img-level (AUROC %) | RESC | 90.9 | 86.2 | 96.2 | 88.8 | 95.3 | 96.1 | 95.2 |
| | HIS | 68.7 | - | 82.7 | 67.5 | 87.5 | 80.6 | 81.6 |
| | ChestXRay | 75.8 | - | 82.0 | 70.0 | 86.7 | 76.4 | 82.7 |
| | OCT | 99.0 | - | 99.4 | 97.9 | 99.1 | 98.9 | 99.8 |
| | BrainMRI | 74.8 | 92.7 | 97.3 | 94.1 | 98.9 | - | 96.5 |
| Pixel-level (AUROC %) | LiverCT | 71.8 | 98.9 | 99.7 | 96.8 | 99.4 | - | 99.5 |
| | RESC | 77.3 | 93.8 | 99.0 | 96.7 | 97.7 | - | 99.0 |
| | Avg. | 77.6 | 88.0 | 92.2 | 82.9 | 94.2 | - | 92.5 |

**Dataset:** For general medical anomaly detection, we follow the BMAD benchmark Bao et al. (2024), which includes 6 datasets: Brain MRI Baid et al. (2021); Bakas et al. (2017); Menze et al. (2014), Liver CT Bilic et al. (2023); Landman et al. (2015), retinal OCT Kermany et al. (2018); Hu et al. (2019), Chest X-ray Wang et al. (2017a), and Digital Histopathology Bejnordi et al. (2017). Among these datasets, both image- and pixel-level metrics are evaluated for BrainMRI Baid et al. (2021); Bakas et al. (2017); Menze et al. (2014), LiverCT Bilic et al. (2023); Landman et al. (2015), and RESC Hu et al. (2019). For the other datasets, namely OCT17 Kermany et al. (2018), ChestXray Wang et al. (2017a) and HIS Bejnordi et al. (2017), only image-level scores are evaluated.

**Experiments:** We first evaluate our method under the setting of general few-shot anomaly detection. We conduct the 4-shot experiments with state-of-the-art few-shot medical anomaly detection methods, MVFA Huang et al. (2024), and other few-shot anomaly detection methods, namely BRA Ding et al. (2022b) and BGAD Yao et al. (2023b). To adapt our method to pixel-level score, we combine our method with MVFA. Specifically, we aggregate our inter-anomaly loss with the losses of MVFA. Following MVFA, we conduct the experiments under the 4-shot setting.

Previous CLIP-based methods, such as WinCLIPJeong et al. (2023) and AnomalyCLIPZhou et al. (2023), were primarily designed for single-anomaly detection. Thus, we can only compare them on the general medical anomaly detection benchmark, which we show as follows. As Table 5 shows, even though our methods are not designed for the general few-shot anomaly detection, we still outperform other methods. The results of WinCLIP are reported by MVFA. Since InCTRL Zhu & Pang (2024) can not output the pixel-level prediction, we only report the image-level results. Note that, during the training stage, AnomalyCLIP requires the full normal and abnormal data and InCTRL requires full normal images, which is different from our few-shot setting in the experiments. Due to the difficulty of accessing medical data, the methods requiring a large amount of data are not suitable for medical anomaly detection in practice. Even with only 4-shot training, our method surpasses the methods requiring much more training data, such as AnomalyCLIP and InCTRL, in some cases, like OCT.

## E    APPENDIX 5: DETAILS OF EXPERIMENTS ON CHESTX-RAY8

We provide the detailed results on ChestX-ray8 under the Anomaly-aware detection protocol in Table 6, which is an extension of Table 1.

Table 6: Experimental results of each anomaly in ChestX-ray8.

| Abnormality | CLIP | MedCLIP | Ours (no SS) | Ours (full model) |
|---|---|---|---|---|
| Atelectasis | 46.6 | 56.1 | 60.9 | 60.9 |
| Fibrosis | 53.6 | 49.0 | 48.2 | 51.2 |
| Mass | 43.5 | 49.2 | 49.3 | 50.7 |
| Infiltration | 50.1 | 54.7 | 52.6 | 49.5 |
| Nodule | 48.6 | 44.1 | 50.4 | 49.7 |
| Effusion | 47.8 | 58.9 | 61.2 | 63.3 |
| Pleural Thickening | 48.0 | 53.2 | 53.3 | 60.6 |
| Pneumothorax | 46.7 | 53.3 | 51.8 | 60.5 |
| Emphysema | 53.2 | 47.6 | 69.1 | 66.8 |
| Cardiomegaly | 55.3 | 50.3 | 55.5 | 55.0 |
| Consolidation | 46.5 | 59.9 | 57.4 | 58.3 |
| Pneumonia | 48.3 | 50.1 | 52.0 | 49.0 |
| Edema | 43.6 | 64.6 | 64.8 | 74.6 |
| Hernia | 44.7 | 42.9 | 57.4 | 55.2 |
| **Avg.** | 48.3 | 52.4 | 56.0 | **57.5** |

## F APPENDIX 6: DETAILS OF EXPERIMENTS ON FASTMRI++

We provide the detailed results on FastMRI+ dataset under the protocol of anomaly-aware detection in Table 7. Since the label *Small vessel chronic white matter ischemic change* can not be achieved in slices 5 and 10, we only evaluate 5 categories within these two slices.

Table 7: The results of the experiments on category-wise AUROC. The reported results are AUROC score (%). We also report average (Avg.) results for each slice. "Small vessel ischemic change" corresponds to the label " Small vessel chronic white matter ischemic change" in the FastMRI+ dataset. "SS" and "Avg." are short for "Sign Selection" and "average" respectively.

| | | CLIP | MedCLIP | Ours(no SS) | Ours(full model) |
|---|---|---|---|---|---|
| | Craniotomy | 42.7 | 50.0 | 68.2 | **70.9** |
| | Posttreatment change | **73.5** | 51.1 | 67.3 | 71.7 |
| | Nonspecific lesion | 56.8 | 44.3 | **65.1** | 56.7 |
| slice 0 | Dural thickening | 44.6 | 48.5 | **58.9** | 57.5 |
| | Enlarged ventricles | 65.3 | 68.7 | 62.5 | **71.9** |
| | Small vessel ischemic change | 62.1 | 39.4 | **81.7** | 80.3 |
| | Avg. | 57.5 | 50.3 | 67.3 | **68.2** |
| | Craniotomy | **67.2** | 46.9 | 55.0 | 55.4 |
| | Posttreatment change | 59.4 | 51.7 | **63.9** | 62.4 |
| | Nonspecific lesion | 47.9 | 44.2 | **64.9** | **64.9** |
| slice 5 | Dural thickening | 51.1 | 63.9 | **64.5** | 57.4 |
| | Enlarged ventricles | 76.1 | 51.3 | 66.6 | **79.3** |
| | Avg. | 60.3 | 51.6 | 63.0 | **63.9** |
| | Craniotomy | 48.3 | 43.0 | 51.6 | **62.4** |
| | Posttreatment change | **61.1** | 58.2 | 40.6 | 46.6 |
| | Nonspecific lesion | 37.5 | 45.6 | **71.3** | 58.0 |
| slice 10 | Dural thickening | 57.7 | 48.8 | **72.2** | 69.9 |
| | Enlarged ventricles | 98.1 | 40.1 | **100** | 70.5 |
| | Avg. | 60.5 | 47.1 | **67.1** | 61.5 |

## G APPENDIX 7: BROADER IMPACTS

Medical anomaly detection has the potential to improve diagnostic accuracy and increase access to quality healthcare significantly. This is particularly critical in resource-constrained environments where access to experienced specialists may be limited.

In this work, we extend conventional anomaly detection frameworks to address multi-anomaly scenarios, where multiple co-existing or interacting abnormalities may be present within a single medical image. Such scenarios more accurately reflect real-world clinical conditions and require models to detect, localize, and differentiate diverse pathological patterns simultaneously.

At a societal level, this advancement can help reduce the diagnostic burden in under-resourced healthcare systems by facilitating earlier and more comprehensive detection of complex diseases. Therefore, we believe our contribution is of substantial value to the medical domain, with the potential to support more informed clinical decisions and promote equitable access to diagnostic tools.

