# OpenReview forum: "SD-MAD: Sign-Driven Few-shot Multi-Anomaly Detection in Medical Images"
_ICLR.cc/2026/Conference — Submitted to ICLR 2026_

### Official Review · Reviewer_SUm2 · 2025-10-31

**Soundness:** 2
**Presentation:** 3
**Contribution:** 2
**Rating:** 4
**Confidence:** 5

**Summary:**

This paper proposes a method for "few-shot" anomaly detection in the medical domain.

**Strengths:**

Pros:
- A notable contribution is that, beyond conventional anomaly detection, the proposed method can also identify anomaly categories.
- The paper is well organized and generally easy to follow.

**Weaknesses:**

Cons:
- There are several unclear or conflicting notations. For instance, in Sec. 3.1, $K$ denotes the number of samples, while in Sec. 3.3, it represents the number of anomaly categories. Similarly, both $i$ and $c$ seem to denote categories. These inconsistencies significantly hinder readability.
- If $K$ indeed refers to the number of anomaly categories, how is it determined? This is not explained in the paper. If it is empirically set, the authors should discuss how its choice influences model performance.
- w/ SS vs. w/o SS. The paper claims that identifying anomaly categories improves performance, yet provides no clear explanation or theoretical reasoning for why this is the case. A deeper discussion or analysis would strengthen the contribution.
- The results are without statistical measures (e.g., mean ± std over 10 trials), which is standard practice in few-shot anomaly detection. This omission weakens the reliability of the comparisons.
- It is unclear how sensitive the model performance is to the selection of samples, which is an important practical consideration.
- The paper lacks any discussion of computational cost or inference time, which are crucial for clinical applicability.
- Why did the authors choose to use the general CLIP model instead of a medical-domain variant? This decision should be justified.
- The paper's definition of "few-shot anomaly detection" seems problematic. In the conventional sense, few-shot anomaly detection extends unsupervised anomaly detection—training with a few normal samples and testing on unseen tasks (which is in line with few-shot learning). However, this work trains on both normal and abnormal samples, which aligns more closely with small sample image classification rather than anomaly detection. The authors should reconsider and clarify the task formulation.

**Questions:**

See Weaknesses.

---

> ### Author Response · Authors · 2025-12-03
>
> We appreciate the reviewer’s constructive comments. Below, we provide detailed responses to each point.
>
> **W1, W2 – Symbol notation:** We appreciate the reviewer for pointing out the notation issue. The numbers of samples and categories are indeed not the same, and the current notation is misleading. We will revise the notation accordingly in the paper.
>
> **W3 – Why we need SS:** As discussed in the paper, SS is designed to address the uncertainty problem in the model. Different symptoms manifest to varying degrees in the images, and their alignment with the textual descriptions also varies. Consequently, some textual descriptions may not be well aligned with their visual representations. To mitigate this issue, we employ SS to filter out outlier textual representations.
>
> **W4, W5 – Sensitivity of the model:** Our experiments are built on the medical AD setting [1], where standard deviations are not reported. For completeness, we report the standard deviation over 3 different seeds, which can be found in the response to **Q1** from **Reviewer 7E6r**
>
> **W6 – Computational cost:** The computational cost is primarily dominated by the vision–language encoder, which is comparable to existing CLIP-based approaches. Since we utilize only a few-shot set of samples per anomaly category, the additional computation introduced by category-specific prompts is minimal.
>
> **W7 – Model choice:** We choose CLIP instead of MedCLIP because chest images constitute the majority of the training data in MedCLIP, which may hinder its ability to generalize to other organs. Our training process requires a model that can generalize across different body parts, such as the brain and eyes.
>
> **W8 – Setting of anomaly detection:** As discussed in MVFA [1], prior methods that make no assumptions about prior abnormal knowledge typically require large numbers of normal samples per class to detect anomalies as deviations from the normal distribution. However, acquiring such large-scale data is impractical in medical anomaly detection. Following the experimental setting of previous works [1,2], our research tackles the challenge of limited known anomalies during training, a scenario that often leads to model bias and poor generalization. To address these challenges, we propose a strategy that enhances the discrepancies between different anomaly categories while mitigating uncertainty within the same anomaly category.
>
> [1] Huang, Chaoqin, et al. "Adapting visual-language models for generalizable anomaly detection in medical images." CVPR 2024.

---

### Official Review · Reviewer_LmK7 · 2025-11-02

**Soundness:** 2
**Presentation:** 3
**Contribution:** 2
**Rating:** 2
**Confidence:** 5

**Summary:**

The paper proposes a CLIP-based few-shot framework for medical abnormality recognition, incorporating (1) radiological “sign prompts,” (2) an inter-anomaly contrastive loss, and (3) a sign selection mechanism at inference. The authors position the work as extending few-shot anomaly detection to multi-anomaly settings and evaluate on ChestX-ray8, OCT-17, and FastMRI+.

**Strengths:**

- Incorporating radiological sign descriptions into visual-language alignment is an interesting idea.
- The few-shot multi-label formulation is practically relevant for real-world medical datasets.

**Weaknesses:**

- The work is not anomaly detection but closed-set multi-label classification. The method assumes all anomaly labels are known during training and requires per-class support images. It does not detect abnormality in an open-set or unsupervised manner, which is the central definition of anomaly detection. The title and claims are therefore misleading.
- No open-set or unseen-class evaluation. There is no experiment for detecting unseen anomaly types. In real clinical anomaly detection, new or rare abnormalities are the norm, yet the paper never addresses this problem. The method collapses to a standard supervised multi-label classifier.
- Pixel-level AUROC is not shown in the main paper, and the appendix combines image-level and pixel-level AUROC into a single arithmetic “average,” which is mathematically invalid and masks the fact that the method trails AnomalyCLIP and MVFA on pixel-level AUROC.
- Strongest baselines (MVFA, AnomalyCLIP) are moved to the appendix and perform comparably or better. The main tables omit the most relevant recent methods, and the appendix shows that the proposed approach does not consistently outperform them. This selective reporting undermines the claimed contribution.
- The radiological sign prompts are not reproducible or clinically validated. The paper does not describe how the signs were generated, who verified them, how many exist per class, or whether they will be released. This prevents scientific reproducibility.
- Lack of evidence that the datasets actually support “multi-anomaly” detection. ChestX-ray8 contains 14 abnormality labels, OCT-17 contains 3, and FastMRI+ contains 6, but the paper provides no statistics on how many anomalies occur per image, nor how often co-occurrence happens. Without reporting label cardinality or multi-label frequency, it is unclear whether the task is truly “multi-anomaly detection” or simply multi-label classification where most samples contain only one abnormality. Since the paper’s motivation depends on co-occurring anomalies, the absence of dataset analysis leaves the main claim unsupported.

**Questions:**

- How does the method behave when a novel anomaly class appears at inference time?
- Why are MVFA and AnomalyCLIP excluded from the main comparison tables?
- Can the authors provide localization maps?
- What percentage of samples in each dataset actually contain more than two anomalies?

---

> ### Author Response · Authors · 2025-12-03
>
> We appreciate the reviewer’s constructive comments. Below, we provide detailed responses to each point.
>
> **W1, W2, Q1 – Exploiting prior abnormal knowledge:** As discussed in MVFA [1], methods that make no assumptions about prior abnormal knowledge typically require a large number of normal samples per class to detect anomalies as deviations from the normal distribution. However, acquiring such large-scale data is impractical in medical anomaly detection. Following the experimental setting of previous works [1,2], our study specifically addresses the challenge of having only limited known anomalies during training, a scenario that often leads to model bias and poor generalization. To tackle these issues, we propose a strategy that enhances the discrepancies between different anomaly categories while mitigating uncertainty within the same anomaly category.
>
> **W3, Q3 – Pixel-level results:** We cannot report pixel-level results in the main experiments because pixel-level annotations are not available in the datasets with multi-category anomalies. Instead, we provide pixel-level results for each dataset under the single-category anomaly detection setting in Table 5 of the appendix.
>
> **W4, Q2 – Limited baseline:**  We include only CLIP and MedCLIP in our main comparison because these are the only existing methods applicable to few-shot multi-label anomaly detection. More details are provided in the response to **Reviewer k1bu** under **W3**. To the best of our knowledge, this is the first work that explicitly focuses on multi-anomaly detection in the medical domain.
>
> **W5 – Generation of sign descriptions:** We use an LLM to generate 15–20 candidate descriptions for each anomaly category. Among these, 3–5 distinct descriptions per anomaly are selected by medical experts. These descriptions are then inserted into different templates, such as “an abnormal photo of {obj} with {anomaly}.” In total, we construct 375 different templates for text generation. We will add these implementation details to the revised manuscript.
>
> **W6 – Existence of multiple anomalies:** As discussed in the experiments, the OCT dataset does not contain co-occurring anomalies. For the other datasets, we provide the statistics below:
> |dataset|number of images with multi-anomaly| number of all images|
> |-|-|-|
> |Chest|2567|16806|
> |fastMRI(selected)|79|293|
>
> [1] Huang, Chaoqin, et al. "Adapting visual-language models for generalizable anomaly detection in medical images." CVPR 2024.

---

### Official Review · Reviewer_7E6r · 2025-11-04

**Soundness:** 3
**Presentation:** 3
**Contribution:** 3
**Rating:** 4
**Confidence:** 3

**Summary:**

The paper introduces SD-MAD, a framework for few-shot multi-anomaly detection (MAD) in medical images. Unlike prior methods treating anomaly detection as a one-class classification problem (normal vs. abnormal), SD-MAD targets scenarios where multiple anomaly categories coexist. The framework is evaluated on diverse datasets using new multi-anomaly metrics, showing clear gains over CLIP and MedCLIP baselines.

**Strengths:**

1.The paper is well written and organized, with clear diagrams (Fig. 1–3) and detailed appendices.

2.It defines and benchmarks few-shot multi-anomaly detection in medical AD area.

**Weaknesses:**

1.Only limited baseline (CLIP and MedCLIP) are compared.

2.The framework relies heavily on manually designed textual signs to represent anomaly semantics. It may make the system sensitive to the quality and accuracy of these textual descriptions. In practice, such signs may contain noise, redundancy, or mismatched terminology relative to the specific medical domain or dataset. When the textual prompts do not align well with the actual imaging characteristics, the performance of SD-MAD may degrade significantly.

3.The novelty of this framework is limited. The core components—prompt-based supervision, adapter tuning, and margin-based loss—are primarily adapted from existing CLIP or few-shot learning paradigms.

**Questions:**

Have the authors examined performance variance across different few-shot sample sets or seeds?

---

> ### Author Response · Authors · 2025-12-03
>
> We appreciate the reviewer’s constructive comments. Below, we provide detailed responses to each point.
>
> **W1-limited baseline:** We include only CLIP and MedCLIP in our main comparison because these are the only existing methods directly applicable to few-shot multi-label anomaly detection. Other approaches, such as AnomalyCLIP, are designed specifically for single-label anomaly detection and cannot be directly adapted to our multi-label setting. For completeness, we additionally provide comparisons with these methods under their original evaluation settings in the appendix.
>
> **W2-textual-quality issue:** We agree that the performance is related to the quality of the textual descriptions, which also motivates us to propose sign selection at inference time to filter out misaligned textual representations. In addition, images are the primary inputs, and in practice, we do not need to generate textual descriptions multiple times for the same image. Therefore, obtaining high-quality text is feasible in real-world settings.
>
> **W3-novelty:** Existing methods and benchmarks for medical anomaly detection are not designed to handle scenarios in which multiple anomalies appear within the same image. As illustrated in Figure 1 of our paper, our proposed setting explicitly considers this more complex and realistic case using a vision–language model. Therefore, we believe our work addresses a meaningful and underexplored challenge and that the proposed research paradigm is computationally efficient.
>
> **Q1-results over different seeds:** We report the results of our full model on the FastMRI slice 5 over 3 different seeds under the setting of integrated anomaly detection:
> ||mean|std|
> |-|-|-|
> |subset acc|22.1|5.2|
> |Hamming score|66.8|9.7|
>
> The mean performance still outperforms both CLIP and MedCLIP.

---

### Official Review · Reviewer_k1bu · 2025-11-08

**Soundness:** 3
**Presentation:** 2
**Contribution:** 3
**Rating:** 6
**Confidence:** 3

**Summary:**

This paper introduces a novel framework for few-shot multi-anomaly detection in medical images. Existing one-class few-shot anomaly detection approaches struggle to distinguish between multiple anomaly categories, which is often critical in real-world clinical scenarios. The authors enhance existing few-shot AD methods by leveraging vision–language alignment, adapting them to handle multi-anomaly detection tasks, and achieving favorable experimental results.

**Strengths:**

1. The paper proposes a few-shot anomaly detection model that can inherently handle multiple anomaly classes within a single framework. By aligning radiological signs with anomaly categories, the method introduces a novel and innovative approach.

2. Experiments conducted across multiple evaluation metrics demonstrate strong and consistent performance gains over baseline methods such as vanilla CLIP, highlighting the effectiveness of the proposed approach. The provided visualizations further corroborate these findings.

**Weaknesses:**

1. The methodology section contains a relatively large number of mathematical formulations, yet several key equations and symbols are insufficiently explained. Clearer definitions and intuitive interpretations would improve readability and reproducibility.

2. The paper devotes substantial space to the Sign Selection process during inference; however, this component appears to have a limited positive effect in certain experiments. The authors should analyze and discuss potential reasons behind this phenomenon.

3. The comparisons in Tables 1–4 are not sufficiently comprehensive. It is unclear why the authors do not include comparisons with CLIP-based approaches, such as AnomalyCLIP, which represent strong and relevant baselines for this task. Incorporating such comparisons would significantly strengthen the validity and credibility of the reported results.

**Questions:**

See Weaknesses

---

> ### Author Response · Authors · 2025-12-03
>
> We thank the reviewer for the constructive comments. Below, we provide detailed responses to each point.
>
> **W1-clearer definition:** We provide a notation table summarizing the key inputs and outputs used in our models:
> |symbol|explanation|
> |-|--|
> |$x_i$|input images|
> |$\textbf{c}_i$|labels corresponding to $x_i$|
> |$c,k$|one anomaly category|
> |$\hat{f}^{in}_i$| input feauture  at $i$-th layer|
> |$\hat{f}^{out}_i$| output feature at $i$-th layer|
> |${f}^{out}_i$| final output at the $i$-th layer mixing the outputs for original layer and adapter layer|
> |$f_{img}^c$| the image feature represetnation corresponding to $c$-th anomaly category|
> |$f_{text,i}^c$| the representation of the $i$-th textual description corresponding the $c$-th anomaly category|
> |$f_{anchor}$ | anchor feature used to determine whether the anomaly category appears; can be used for binary anomaly detection.|
>
> We will correct the misnotated symbols and provide a more detailed and comprehensive notation table in the revised manuscript.
>
> **W2-reason for limited positive effect:** As discussed in the last section of our paper, different anomalies may share the same textual descriptions. For instance, “surgical resection” is a description associated with Craniotomy, but Post-treatment Change may exhibit the same characteristic. This ambiguity can lead to failure cases and limited improvement in some scenarios. Moreover, if the text features are already well separated in the embedding space, the proposed selection strategy is expected to yield only marginal additional gains.
>
> **W3-limited CLIP-based methods:**  We include only CLIP and MedCLIP in our main comparison because these are the only existing methods directly applicable to few-shot multi-label anomaly detection. Other approaches, such as AnomalyCLIP, are specifically designed for single-label anomaly detection and cannot be straightforwardly adapted to our multi-label setting. For completeness, we additionally provide comparisons with these methods under their original evaluation settings in the appendix.

---

### Meta-Review · Area_Chair_oQaT · 2025-12-14

**Summary:**

The paper proposes a new approach for few-shot medical anomaly detection in a challenging scenario where a set of multiple
anomaly categories may present in one image input. The method uses diverse text prompts to characterize the anomaly categories and then align the image embeddings with these anomaly-specific text prompts during training. It shows effective detection performance on  three medical imaging datasets.

**Reviewer Concerns:**

The work receives four reviews, with a wide range of concerns. The major concerns are summarized as follows:
- **Research motivation.** The research problem is considered as closed-set supervised multi-label classification, rather than an anomaly detection task as claimed in the work. Anomaly detection is inherently an open-set or unsupervised learning problem. In addition, the work also does not show statistics on image-wise anomaly co-occurrence, undermining the claimed “multi-anomaly” scenarios.
- **Technical design and novelty.** Key designs such as sign selection offer limited performance gains in part of the results. The framework offers limited novelty, as its main components (prompt supervision, adapter tuning, margin-based loss) are largely adapted from existing CLIP or few-shot learning paradigms.
- **Theoretical/Empirical justification.** Baseline comparisons omit strong and closely relevant methods such as AnomalyCLIP, MVFA, and other recent CLIP-based anomaly frameworks. Important design choices, e.g., the number of anomaly categories and the selection of support samples, are not justified or evaluated. The model’s sensitivity to textual sign quality is also not analyzed despite its significant role in the proposed approach.

For the AD setting, the author rebuttal provides justification for why labeled anomaly data is used, but it is unclear why the authors do not evaluate the method in realistic AD settings where unknown/unseen anomalies are always expected.

As for the experiment comparison, the AC is not convinced by the rebuttal arguments, such as "*existing methods are designed specifically for single-label anomaly detection and cannot be directly adapted to our multi-label setting*...", "*As discussed in MVFA [1], methods that make no assumptions about prior abnormal knowledge typically require a large number of normal samples per class to detect anomalies as deviations from the normal distribution. However, acquiring such large-scale data is impractical in medical anomaly detection.*" and a few other similar statements. This is because these single-label AD methods could be rather easily adapted to the proposed setting, and recent AD methods show strong zero-shot or few-shot AD performance without relying on large labeled data in the target applications.

The AC also agrees with the reviewers that the motivation and the advantages of the proposed model designs require more in-depth investigation.

**Reviewer Scores:**

The paper receives one weak accept, two weak rejects, and one reject. Although the rebuttal helps address some of the concerns, such as paper clarity and computational overhead, the key concerns over problem setting, experiments, and technical contributions require further significant investigation.

---

### Decision · Program_Chairs · 2026-01-26

Reject